

# Mouse spermatozoa with higher fertilization rates have thinner nuclei

Daisuke Mashiko[1], Masahito Ikawa[2] and Koichi Fujimoto[1]

[1] Department of Biological Sciences, Osaka University, Toyonaka, Osaka, Japan
[2] Research Institute for Microbial Diseases, Osaka University, Suita, Osaka, Japan

## ABSTRACT

**Background**. Although spermatozoa with normal morphology are assumed to have uniform fertilization ability, recent data show that even normal spermatozoa have considerable variation in their head shape which is associated with differences in fertilization ability. Appropriate quantitative indicators for good sperm morphology, however, remain unidentified.

**Methods**. Therefore, in an effort to identify such an indicator, we compared the nuclear contour of normal mouse spermatozoa by quantitative multivariate analysis using elliptic Fourier descriptors combined with principal component analysis. The spermatozoa were obtained from different strains and collection sites which have been shown to be associated with different fertilization abilities.

**Results**. We found that the head was 5.7% thinner in spermatozoa from the B6D2F1 (BDF1) strain, known to have a higher fertilization rate, than in those from the C57BL/6N (B6N) strain, which has a lower fertilization rate. Moreover, zona-penetrated spermatozoa in the perivitelline space consistently had 5.4% thinner heads than those isolated from the epididymis before ejaculation. The aspect ratio, which represents the sperm head thinness, uniquely distinguished these sperm populations, confirming its validity as a morphological indicator.

**Discussion**. Because aspect ratio has also been shown to characterize human spermatozoa, this unique morphometric indicator might be applicable to compare normal spermatozoa among multiple patients, which will greatly facilitate and enhance current reproductive technologies.

## INTRODUCTION

Male infertility causes half of infertility cases in humans (*Irvine, 1998*). To assess male infertility, features of the population of spermatozoa in the semen that positively correlate with fertilization ability, including the ratio of spermatozoa showing normal morphology and the number of motile spermatozoa (*World Health Organization, 2010*), have been used (*Jouannet et al., 1988*; *Toner et al., 1995*; *Eggert-Kruse et al., 1996*; *Coetzee, Kruger & Lombard, 1998*; *Menkveld et al., 1990*; *Van Waart et al., 2001*; *Garrett et al., 2003*; *Liu, Garrett & Baker, 2003*). One feature that can be assessed in a low-cost, high-throughput manner is sperm head morphology, which is affected by defects in nuclear structures (DNA (*Gandini et al., 2000*), chromosomes (*Lee, Kamiguchi & Yanagimachi, 1996*), chromatin

Corresponding authors
Daisuke Mashiko,
mashiko@bio.sci.osaka-u.ac.jp
Koichi Fujimoto,
fujimoto@bio.sci.osaka-u.ac.jp

(*Dadoune, Mayaux & Guihard-Moscato, 1988*; *Devillard et al., 2002*; *Martin et al., 2003*)).
Most early studies on sperm fertilization indicators distinguished normal sperm from
abnormal sperm that failed to fertilize or resulted in embryos that spontaneously aborted
(*Fredricsson & Björk, 1977*; *Menkveld et al., 1990*; *Menkveld et al., 1991*; *Liu & Baker, 1992*;
*Vilyana et al., 2017*). However, it would be beneficial to distinguish sperm with a better
chance of fertilization among the population of normal sperm. For example, selection of
good sperm morphology is beneficial when injecting a single spermatozoon into an ovum
in the process known as intracytoplasmic sperm injection (ICSI) (*Palermo et al., 1992*) to
resolve male infertility. It is unclear whether good sperm can be identified by indicators of
infertility or whether they are represented by other morphologies, because normal sperm
indicators that are quantitatively correlated with the success of fertilization have not been
well studied.

Even normal spermatozoa in semen are morphologically heterogeneous in their
head shape in mammals (*Severa et al., 2010*; *Ramón et al., 2014*); human normal sperm
morphologies have a range of 2.5–3.2 µm in width and 3.7–4.7 µm in length based on
the WHO manual (*World Health Organization, 2010*). Interestingly, semen containing an
abundance of normal spermatozoa with elongated heads show higher fertilization ratio
than that with rounded heads (*Ramón et al., 2014*). This comparative morphology of sperm
achieved some success as it was used to arbitrarily classify continuous variations of sperm
head morphologies into several discrete categories which correlate fertilization ratio.

To distinguish normal sperm head morphologies, it has been unclear whether
conventionally used indicators such as length, width, and angle are optimal. Moreover,
they could underestimate complex head contours by merely measuring distances
between subjectively selected points (Fig. S1 in Supplemental Information). Geometric
morphometrics combining multivariate quantification of contour with systematic
extraction of indicators have been developed to circumvent such shortcomings, and
this analysis has been applied to evaluate morphological heterogeneities in mice (*Oka
et al., 2007*), bovine (*Ostermeier et al., 2001*), stallion (*Severa et al., 2010*), and human
(*Utsuno et al., 2013*) sperm. The Elliptic Fourier descriptor (EFD) is a representative
method to quantify any contour (closed curve) using multiple ellipses (*Severa et al.,
2010*; *Kuhl & Giardina, 1982*; *Iwata & Ukai, 2002*; *Utsuno et al., 2013*; *Beletti, Da Fontoura
Costa & Viana, 2005*). Principal component analysis (PCA) can provide the optimal
indicators to maximize variance among data (head contour of spermatozoa) and compress
multivariate information into lower dimensions while retaining most of the original
information. Therefore, a pipeline integrating geometric morphometrics with comparative
morphology among normal spermatozoa with different fertilizability could objectively
identify morphological indicators that correlate with the success of fertilization.

To develop a pipeline that can determine indicators for high fertilizability, mice
spermatozoa are a valuable model system because spermatozoa having known differences
in fertilization rates (i.e., collection sites of spermatozoa and mouse strains) are available.
The hybrid BDF1 strain (F1 of inbred B6N × DBA/2) has a higher fertilization ability than
the inbred C57BL/6N (B6N) strain (*Sztein, Farley & Mobraaten, 2000*). Spermatozoa that
reach close proximity to the egg (i.e., those in the cervical canal and in oviduct) are known

to have higher fertilizability than those collected just after ejaculation (*Sztein, Farley & Mobraaten, 2000*; *Cohen & McNaughton, 1974*; *Fischer & Adams, 1981*; *Siddiquey & Cohen, 1982*). Therefore, we applied geometric morphometrics to analyze sperm head contours within a population and between mouse strains and/or collection sites of spermatozoa. To identify morphological indicators of highly fertile spermatozoa, and to develop a pipeline that might also be applicable to human normal sperm.

## MATERIALS AND METHODS

### Animals

All animal experiments were approved by the Animal Care and Use Committee at the Research Institute for Microbial Diseases at Osaka University [permit number: 2589, 3514]. C57BL/6N (B6N) and BDF1 mice (12 weeks old) were obtained from SLC (JapanSLC, Inc. Shizuoka, Japan).

### Abnormality test

Based on the morphological criteria for sperm abnormality that were described previously (*Touré et al., 2004*; *Wyrobek & Bruce, 1975*; *Bruce, Furrer & Wyrobek, 1974*; *Watanabe & Endo, 1991*), we defined abnormal sperm as size outliers that showed a PC1 score > 0.55 and largely lacked the hook-shaped heads common to most sperm.

### *In vitro* fertilization and spermatozoon collection

Females ($N = 16$) were superovulated by injecting 5 IU of pregnant mare serum gonadotropin (PMSG, ASKA Pharmaceutical Co., Ltd. Tokyo, Japan), and then, 5 IU of human chorionic gonadotropin (HCG, ASKA Pharmaceutical Co., Ltd. Tokyo, Japan) 48 h after the PMSG injection. The ovulated oocytes were collected from the oviducts 14 h after the HCG injection. Cumulus-enclosed oocytes were placed in 100 μl drops of TYH medium (*Toyoda, Yokoyama & Hoshi, 1971*) covered with paraffin oil (Nacalai Tesque, Kyoto, Japan). The spermatozoa collected by mechanically-dissecting cauda epididymides ($N = 5$) were placed in 100 μl drop of TYH medium. After a 2-h incubation, the sperm suspension in TYH was added to the TYH drop containing eggs at a concentration of $5 \times 10^5$ sperm/ml. After 2–8 h co-insemination of the spermatozoa and oocytes, the zona-bound spermatozoa were carefully removed using a holding needle. Because of the technical difficulty associated with collecting sperm in the zona pellucida of an oocyte by manipulation (*Inoue et al., 2011*), we obtained zona-penetrated spermatozoa using acidic Tyrode's solution after *in vitro* fertilization. Acidic Tyrode's solution treatment did not affect the sperm head aspect ratio. Zona pellucida was removed from the oocyte in 20 μl acidic Tyrode's solution drops on a glass slide. The spermatozoa attached to the egg surface were removed by pipetting, and subsequently, the eggs were removed from the drops.

Epididymal spermatozoa were collected by dissecting the caput, corpus and cauda epididymides ($N = 5$ for each experiment) and placed into 400 μl phosphate buffered saline (PBS). Spermatozoa were collected from the oviduct and uterus by flushing out these structures with PBS 4 h after coitus.

## Spermatozoon imaging and analysis

The spermatozoa were coated onto a glass slide (Matsunami Glass, Osaka, Japan) and stained with 65 μM Hoechst 33342 (Life Technologies, Carlsbad, CA, USA). The slides were viewed using an Olympus IX-70 fluorescence microscope with a 10× eyepiece and 100× objective lens (Numerical aperture is 1.4). The sperm head contours were derived from image binarization using the discriminant analysis method (*Otsu, 1975*) in openCV (available also in ImageJ, http://imagej.nih.gov/ij/), which was customized using the C programing language.

## Elliptic Fourier descriptors and principal component analysis

Each sperm head was transferred to EFDs with two-dimensional coordinates given by:

$$
\begin{cases}
X(t) = \sum_{n=1}^{N} \left( a_n \cos \frac{2n\pi t}{T} + b_n \sin \frac{2n\pi t}{T} \right) \\
Y(t) = \sum_{n=1}^{N} \left( c_n \sin \frac{2n\pi t}{T} + d_n \cos \frac{2n\pi t}{T} \right)
\end{cases}
\tag{1}
$$

where $n$, $N$, $t$, and $T$ denoted the harmonic number, the maximum harmonic number, the displacement along the contour, and the total displacement, respectively. At $N = 1$, $(X(t), Y(t))$ represented an ellipse. We set $a_n$, $b_n$, $c_n$, and $d_n$ as parameters of the PCs. The number of coefficients was provided as $4N\text{-}3$ because normalization was carried out for the size and angle of the first harmonic ellipse with $a_1 = 1$, $b_1 = 0$, and $c_1 = 0$. The variable $d_1$ denoted the aspect ratio. We approximated the contours of the heads of spermatozoa up to 20 ellipses ($N = 20$), and we performed the PCA on 77 parameters and the data reconstruction on the PC scores using SHAPE (http://lbm.ab.a.u-tokyo.ac.jp/~iwata/shape/index.html) (*Iwata & Ukai, 2002*).

## Statistics

Kolmogorov–Smirnov test, Shapiro–Wilk test for checking normality, $F$-tests for checking homoscedasticity, $t$-tests and Steel-Dwass tests were performed using custom R programs. A $P$-value > 0.05 was considered not significant (n.s.), whereas $P$-values < 0.05 (*), <0.01 (**), and <0.001 (***) were considered significant.

# RESULTS

## EFD and PCA revealed sperm head aspect ratio as unique fertility indicator

In order to compare zona penetrated spermatozoa and ejaculated sperm, we focused on contour of the nucleus, which does not change by spontaneous acrosome reaction (See Figs. S2A–S2C). To quantify the variations in sperm head morphology, we first collected spermatozoa ($n = 179$) from dissected B6N male cauda epididymides (Fig. 1A) and tracked the sperm head contour (Fig. 1B) by taking the pictures of each spermatozoon nucleus (Fig. S3). The tracked contour was sequentially input into a quantitative descriptor EFD method (see "Image analysis" and "Elliptic Fourier descriptors and principal component analysis" in 'Materials and Methods'; Fig. S4). To extract normal sperm characteristics,

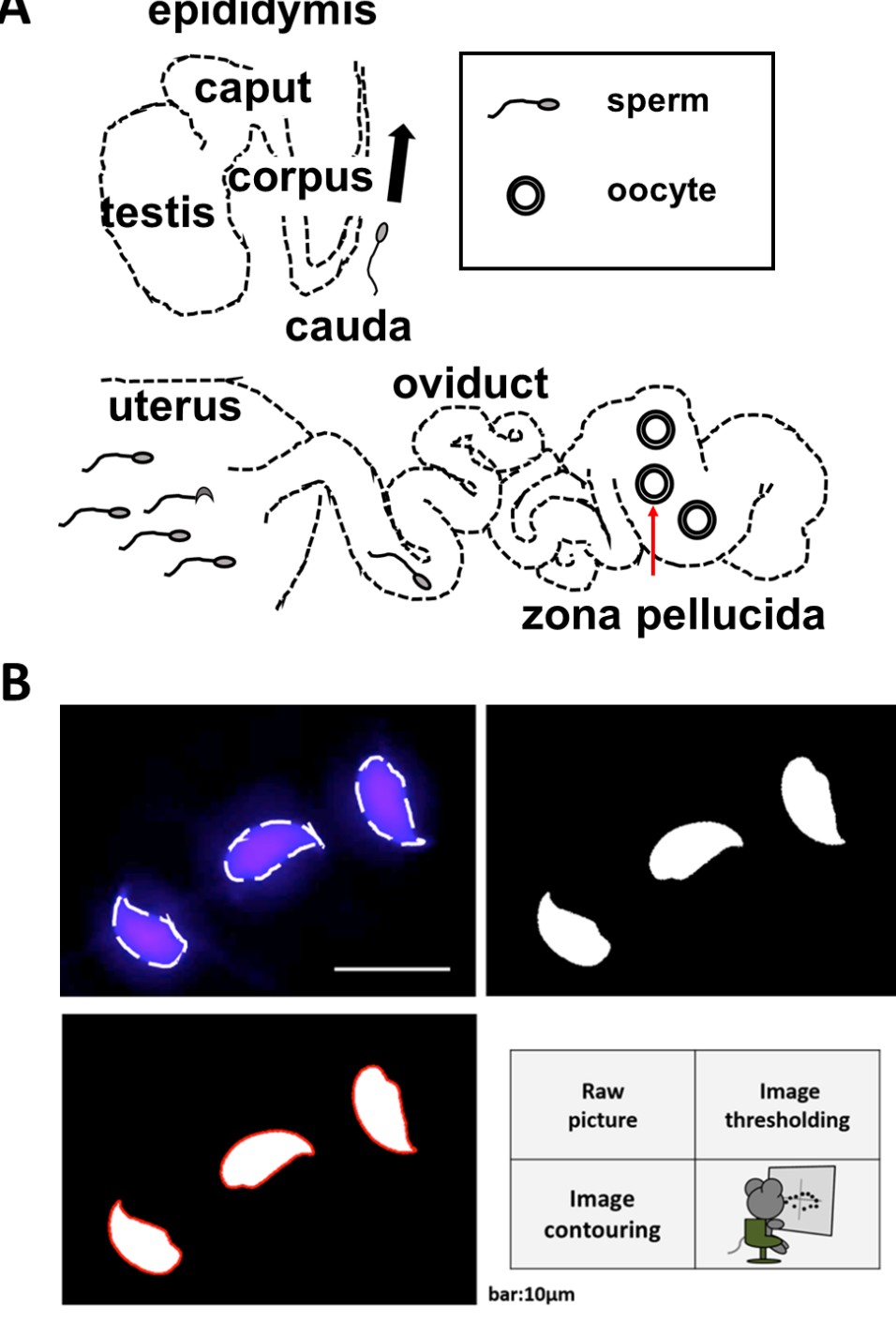

**Figure 1  Schematic diagrams illustrate the methods of spermatozoon isolation and sperm head contour extraction.** (A) The male and female reproductive tracts. Spermatozoa were collected from the epididymis of males or the oviducts or uteri of females. Zona-penetrated spermatozoa were isolated following *in vitro* fertilization. The red arrow shows the zona pellucida. (B) The shapes of the sperm heads were visualized using Hoechst 33342 to stain the nuclei. Image binarization and extraction of the contours were subsequently performed.

we subsequently performed PCA after subtracting out the abnormal spermatozoa (see "Abnormality test" in 'Materials and Methods'; Fig. S5). Applying this protocol, we quantified the variation in head morphology of normal spermatozoa, as optimally separated into multiple principal components (PCs; Fig. S6), among B6N ($n = 170$, Fig. 2A) and BDF1 spermatozoa ($n = 163$, Fig. 2B). PC1 of the B6N spermatozoa highlighted variations in width (Fig. 2A and S6), whereas PC2 highlighted variations in the hook shape of the tip (Fig. S6).

To identify the PCs that correlate with fertilization ability, we focused on PC1, PC2, and PC3, whose contribution rates were above 10% for B6N normal sperm (49.8%, 19.3%, and 12.7%, respectively; Fig. S6). First, we identified the PCs that distinguished the BDF1 from the B6N spermatozoa and the epididymis-isolated from the zona-penetrated B6N spermatozoa (see "*In vitro* fertilization" in 'Materials and Methods'). Regarding mouse strain, the BDF1 and B6N spermatozoa collected from the cauda epididymis showed differences in PC1, PC2, and PC3 (Fig. 2C and S7). However, regarding spermatozoa from different collection sites of spermatozoa, the epididymis-isolated and zona-penetrated B6N spermatozoa differed in PC1, but not in PC2 or PC3 (Fig. 2C and S7). Thus, PC1 uniquely distinguished spermatozoa between mouse strains and between collection sites of spermatozoa.

Next, we tried to identify the morphological parameters contributing to PC1. The width of the sperm head, defined as the longest part of the head perpendicular to its length, seemed to be highlighted in PC1 (Fig. 2A), but the error in width measurement was large. On the other hand, the major and minor axes of the ellipse representing the lowest mode of the sperm head EFD were almost precisely overlapped with the antero-posterior (length) and dorso-ventral (width) axes of the sperm head, respectively (Fig. 3A and Fig. S8A). Thus, we used these major and minor axes to calculate the aspect ratios (minor axis divided by the major axis, equivalent to $d_1$ in Eq. (1); Fig. 3A). We found that the sperm head aspect ratios of BDF1 spermatozoa ($n = 330$) were reduced compared with those of B6N spermatozoa (Fig. 3B; $n = 298$; one-tailed $t$-test, $P = 2.1 \times 10^{-16}$), whereas the sperm head areas were not significantly different between the two groups (Figs. S9A and S9B; $n = 687$ for BDF1; $n = 465$ for B6N; two-tailed $t$-test, $P = 0.53$). Moreover, the factor loading of the aspect ratio on PC1, which was proportional to the correlation coefficient with PC1, was more than 2-fold higher than that of any of the other parameters of the EFDs ($x$-coordinate in Fig. 3C). Thus, by combining PCA and EFDs, we established a quantitative evaluation pipeline of multi-dimensional morphological variation, and using this pipeline, we identified a low aspect ratio, i.e., a thin sperm head, as a unique and optimal morphological indicator of the fertilization ability of spermatozoa.

## The sperm head aspect ratio decreased during the progression of fertilization

We next compared the sperm head aspect ratio from cauda epididymis-isolated B6N spermatozoa ($n = 118$, green in Fig. 4) with zona-penetrated B6N spermatozoa isolated from the perivitelline space (PVS) ($n = 133$, red in Fig. 4 and Fig. S10). The sperm head aspect ratio of zona penetrated spermatozoa was significantly smaller than that of cauda

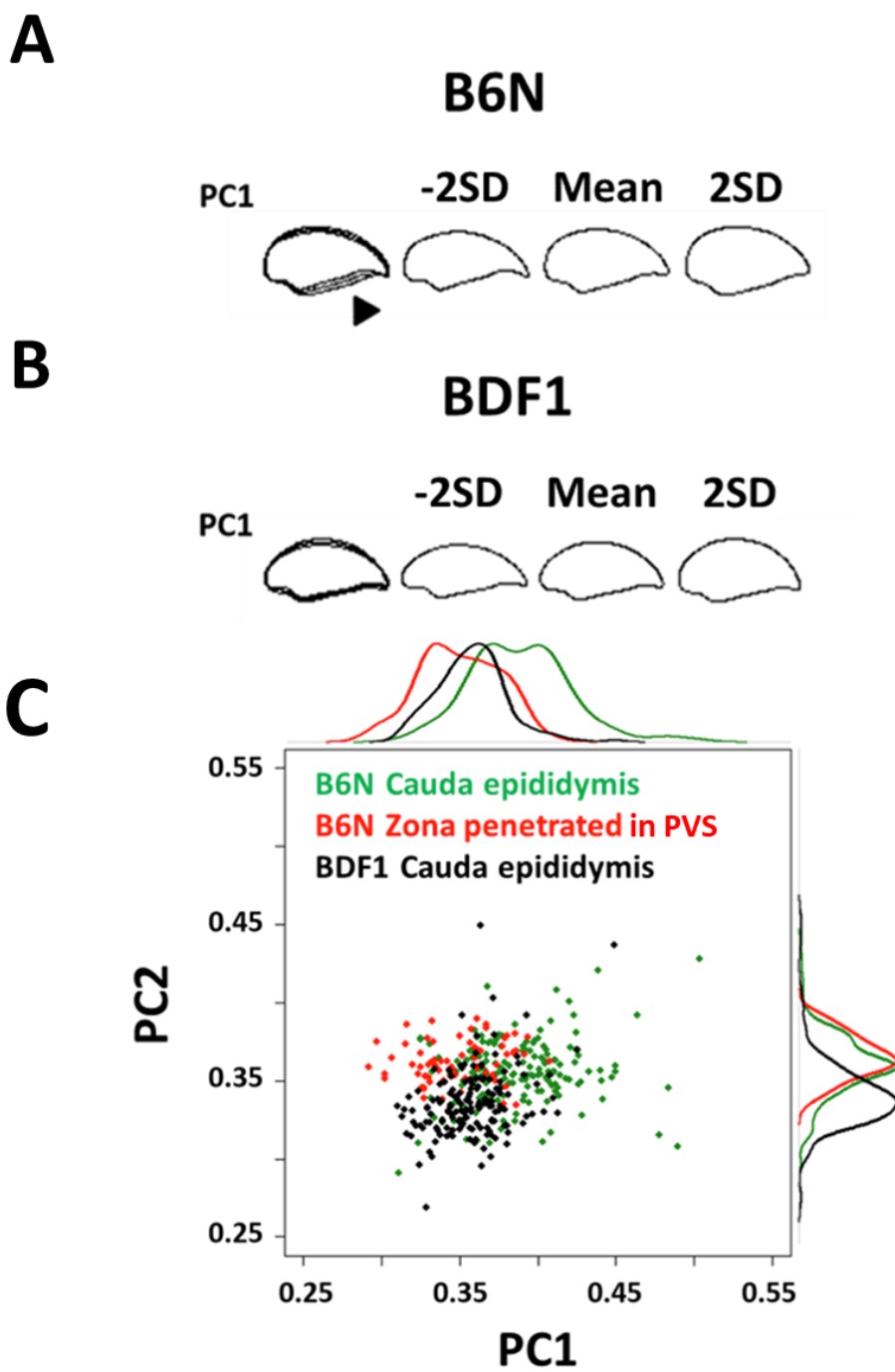

**Figure 2 The first principal component distinguished variation in normal spermatozoa among strains and fertilization stages.** Minus 2 standard deviation (SD), mean, and +2 SD sperm head contours of the first principal component (PC1) are shown from (A) B6N and (B) BDF1 mouse epididymis-isolated spermatozoa. The arrowhead shows the region of increased variation. (C) A scatter plot and the associated density curves (outside of the $x$- and $y$-axis) of the PC scores of B6N zona-penetrated spermatozoa (red), and B6N (green) and BDF1 (black) cauda epididymis-isolated spermatozoa. The distribution of each group partially overlapped at PC1 = 0.3–0.4. The eigenvectors of the PCs were derived from the principal component analysis (PCA) of B6N epididymis-isolated spermatozoa.

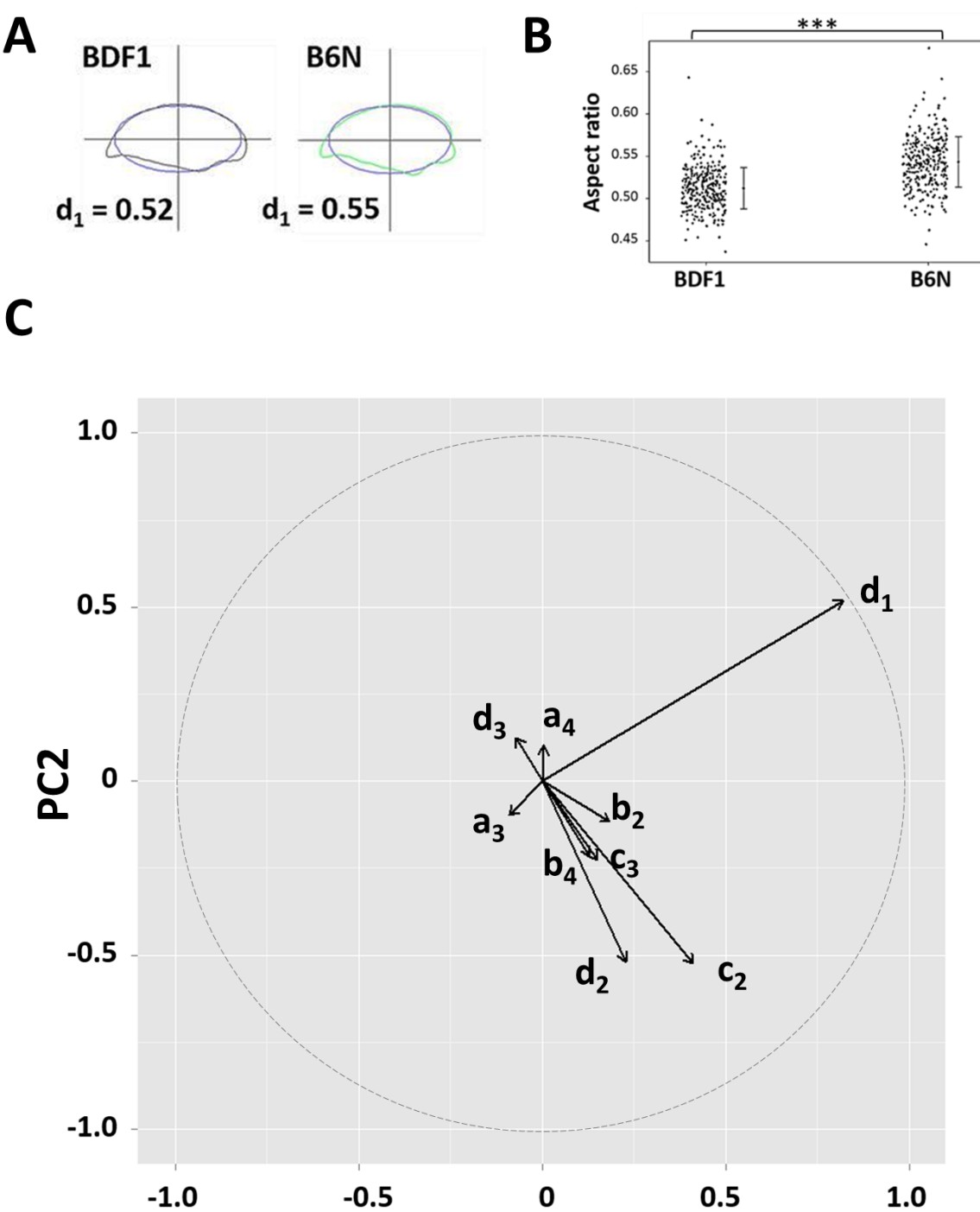

**Figure 3  The sperm head aspect ratio was the largest contributor to the first principal component.** (A) Representative sperm head shapes of BDF1 (gray) and B6N (green) epididymis-isolated spermatozoa are shown with the overlapping ellipse (blue) used to calculate population mean of the aspect ratio ($d_1$ in Eq. (1)). (B) A dot plot shows the aspect ratio of the heads from BDF1 and B6N spermatozoa. The error bars denote the standard deviation (SD). (C) The arrows show the eigenvectors of the PC projected into PC1-PC2 space, whose length (the square-root of the sum of the square of the factor loadings of PC1 and PC2) is greater than 0.1, and where the dashed circle denotes the norm of the eigenvector $= 1$.

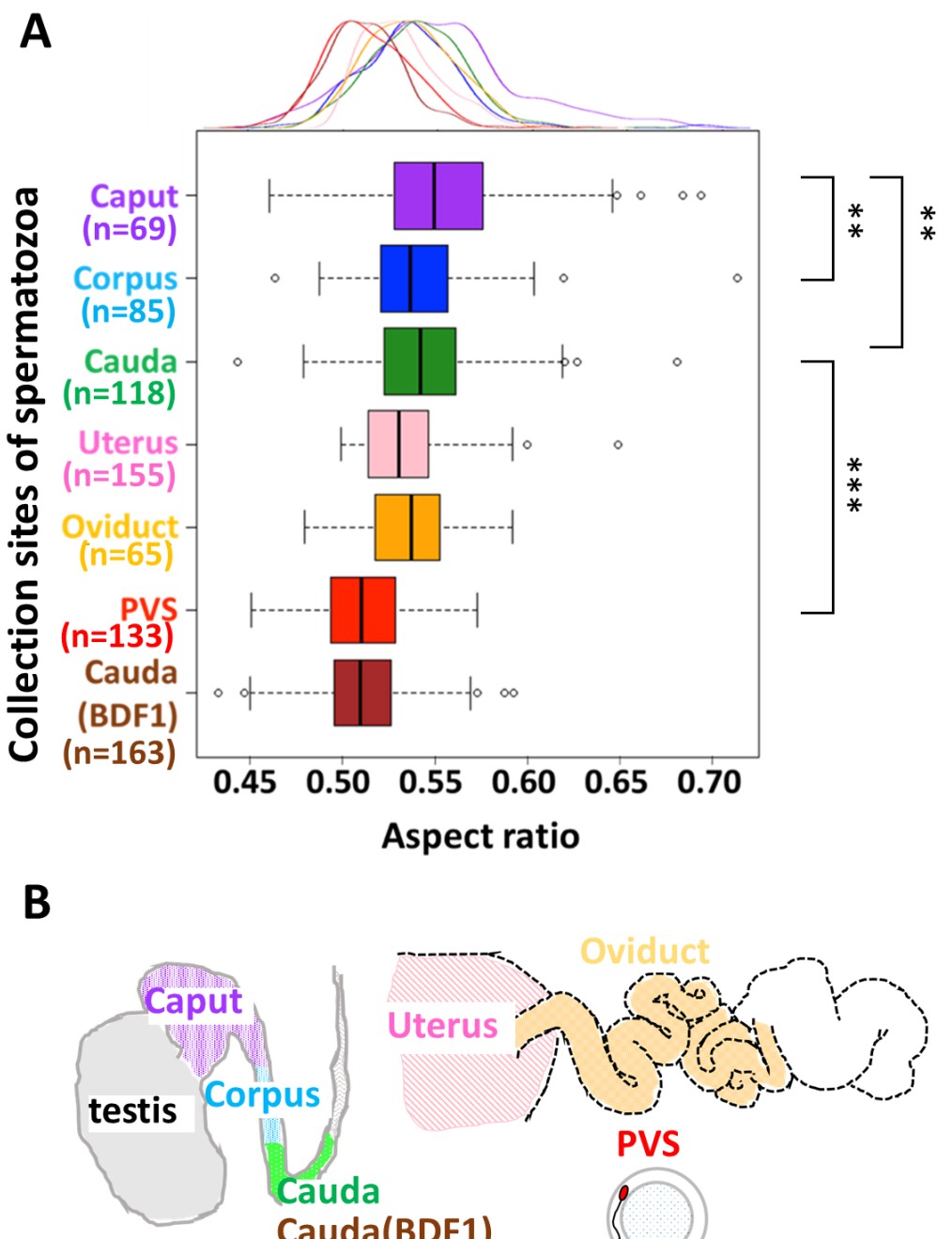

**Figure 4** **The sperm head aspect ratio decreased during fertilization.** (A) A box plot, with a corresponding density plot above the graph, depicts the sperm head aspect ratios, where the left/right hinge and the thick middle line represent the 25th/75th and 50th percentile, respectively. Spermatozoa were collected from the caput (purple), corpus (sky blue), and cauda (green) regions of the epididymis from B6N male mice ($N = 5$), or from the cauda epididymis from BDF1 male mice (brown; $N = 5$). B6N spermatozoa were also recovered from the uterus (pink) or oviducts (orange) of female mice ($N = 16$), and zona penetrated spermatozoa were recovered from the perivitelline space (PVS) of oocytes (red). The variance of the aspect ratio did not change during zona penetration (green and red; $F$-test, $P = 0.12$).(B) The male and female reproductive tracts.

spermatozoa (Fig. 4; unpaired, one-tailed $t$-test, $P = 2.2 \times 10^{-16}$), further confirming the validity of the sperm head aspect ratio as a fertilization indicator. Moreover, the B6N zona penetrated spermatozoa had an aspect ratio that was similar to that of BDF1 cauda epididymis-isolated spermatozoa (Fig. 4, Kolmogorov–Smirnov test, $P = 0.57$), which have a higher fertilization rate than B6N epididymis-isolated spermatozoa. Importantly, the mean aspect ratio of spermatozoa decreased throughout the progression of fertilization, and it was consistently negatively correlated with the fertilization ability of the spermatozoa. To investigate whether the sperm head aspect ratio decreased during sperm maturation in male mice prior to ejaculation, we collected spermatozoa from each region of the epididymis, including the caput, corpus, and cauda (Fig. 1A). The mean aspect ratio of spermatozoa from the caput ($n = 69$, purple box in Fig. 4) was 3.0% larger than that from the corpus ($n = 85$, sky blue in Fig. 4; Steel-Dwass test, $P = 0.01$) and 2.6% larger than that from the cauda ($n = 118$, green in Fig. 4; Steel-Dwass test, $P = 0.008$); however, the mean aspect ratios of spermatozoa from the corpus and cauda were not different from each other (Steel-Dwass test, $P = 0.98$). In addition, the minor axis length of the ellipse was decreased by 3.1% from the caput to the cauda (Fig. S8B; two-tailed $t$-test, $P = 0.015$), whereas the major axis length was not significantly different between those groups (two-tailed $t$-test, $P = 0.58$). Similar to the aspect ratio means, the coefficients of variation (i.e., SD normalized by mean) of the aspect ratios of B6N spermatozoa continuously decreased from the caput (8.4%), to the corpus (6.0%), to the cauda (5.3%).

Finally, we also collected B6N spermatozoa from the uteri ($n = 155$) and the oviducts ($n = 65$; Fig. 1A) of females post-coitus, and we did not observe a difference in mean aspect ratio between these two locations (unpaired two-tailed $t$-test, $P = 0.22$). Moreover, to address whether the acrosome reaction (*Dan, 1952*) causes the decreased aspect ratio of the zona-penetrated spermatozoa, we analyzed sperm morphology before and after the acrosome reaction by using transgenic Acr-EGFP (EGFP fused to the acrosome migrating signal; Fig. S2) mice and found that the acrosome reaction did not affect the aspect ratio in sperm nucleous. Taken together, these data demonstrate that the aspect ratio of the sperm head did not change during entry of spermatozoa into the oviduct, but decreased during spermatozoon zona penetration.

## DISCUSSION

### Possible mechanisms of the decreasing sperm head aspect ratio throughout the fertilization process

During spermatozoon maturation in the epididymides of B6N male mice, we found major morphological changes including decreases in head aspect ratios and minor axis lengths of the normal spermatozoa (Fig. 4 and Fig. S8B). These decreases are likely due to structural changes that take place inside the sperm nucleus, which largely occupies the sperm head. During spermatogenesis in the testis, the DNA packaging histones are replaced by protamines, which more densely package every 50–60 kb of DNA into a toroid (donut) shape (*Braun, 2001*). In the caput, the toroids are subsequently cross-linked by disulfide (SS) bonds, resulting in further DNA compaction (*Bedford & Calvin, 1974*), which

is consistent with our observed decrease in the sperm head minor axis length and aspect ratio in the caput-isolated spermatozoa. Thus, we hypothesize that the decreased aspect ratio in maturing spermatozoa from the epididymis is caused by the formation of SS bonds, which can be tested using dithiothreitol to reduce the SS bonds (*Bedford & Calvin, 1974*) and determine whether they are indeed responsible for these morphological changes.

In the female reproductive tract, the ratio of abnormal to normal spermatozoa was previously reported to decrease considerably at the utero-tubal junction (UTJ; junction of uterine and oviduct) (*Krzanowska, 1974*; *Nestor & Handel, 1984*). Here, we showed that the normal spermatozoa isolated from the uteri and oviducts after coitus had similar sperm head aspect ratios (Fig. 4), suggesting that UTJ plays a role in eliminating the abnormal spermatozoa but not in altering normal sperm morphology. The decreases in sperm head aspect ratios that we observed during zona penetration (Fig. 4) could be due to the selection of a fractional sperm population (*Utsuno et al., 2013*; *Beletti, Da Fontoura Costa & Viana, 2005*) or to the deformation of individual sperm. Sperm deformation has been attributed to the acrosome reaction (*Dan, 1952*), which, in our case, we showed to be unlikely (Fig. S2), or to the mechanical force loaded at the time of the zona penetration. Thus, determining whether sperm subpopulation selection or mechanical deformation occurs during zona penetration, using methods such as time-lapse imaging and/or measuring the yield stress on the sperm head during zona penetration, will be important avenues for future examination.

### Strategies for identifying the most fertile spermatozoa and implication for therapeutic applications

The correlation of sperm head morphology and fertilization ability (Figs. 2C and 4) suggests that spermatozoon morphology can be used as a strategy to screen mouse spermatozoa suitable for fertilization. Here, we showed that populations of spermatozoa with higher fertilization abilities (BDF1 cauda and B6N zona-penetrated in Fig. 4) had smaller aspect ratios, i.e., thinner heads, than other populations. The generality of this correlation should be further validated by applying our morphometry to zona-penetrated spermatozoa of other strains whose fertilization rates are well known, including DBA/2, BALB/c, 129S3/SvIm, and FVB/N. Moreover, even in spermatozoon populations with a large mean aspect ratio, there could be subpopulations with smaller aspect ratios and increased fertilization abilities. For instance, in our study, we observed a fraction of B6N cauda spermatozoa that morphologically overlapped with BDF1 cauda spermatozoa and B6N zona-penetrated spermatozoa, which have higher fertilization abilities (PC1 = 0.3–0.4 in Fig. 2C). Thus, evaluating whether these morphologically unique subpopulations have correspondingly altered fertilization abilities will be another important avenue for future examination that will ultimately enable us to screen heterogeneous spermatozoon populations to identify spermatozoa that are suitable for fertilization by their morphological characteristics, such as their sperm head aspect ratio.

### Applicability to human sperm

The correlation between aspect ratio and the success of fertilization in mice is consistent with a study showing that elongated sperm is more favorable for fertilization than rounded sperm

(*Ramón et al., 2014*). Human spermatozoa have been represented mainly by ellipses using Elliptic Fourier descriptors (*Utsuno et al., 2013*). Therefore, the present pipeline, which integrates geometric morphometrics with comparative morphology (Figs. 2 and 3), could be applied under microscope in clinic laboratories, as long as edge detection of multiple spermatozoa is performed with sufficiently low error rate (Fig. 1B) so as to quantitatively examine the correlation between the aspect ratio normal sperm and fertilization success among human patients. This would greatly facilitate and enhance current reproductive technologies in an objective, high-throughput manner.

## ACKNOWLEDGEMENTS

We thank S Nishioka and Y Esaki for preparing the zygotes; members of the theoretical biology laboratory for constructive criticisms; SAM Young and T Yao for critical reading of the manuscript.

### Funding
Daisuke Mashiko is a JSPS Research Fellow (17J01156). The funders had no role in study design, data collection and analysis, decision to publish, or preparation of the manuscript.

### Grant Disclosures
The following grant information was disclosed by the authors:
JSPS Research Fellow: 17J01156.

### Competing Interests
The authors declare there are no competing interests.

### Author Contributions
- Daisuke Mashiko conceived and designed the experiments, performed the experiments, analyzed the data, wrote the paper, prepared figures and/or tables, reviewed drafts of the paper.
- Masahito Ikawa contributed reagents/materials/analysis tools, reviewed drafts of the paper.
- Koichi Fujimoto analyzed the data, wrote the paper, reviewed drafts of the paper.

### Animal Ethics
The following information was supplied relating to ethical approvals (i.e., approving body and any reference numbers):

All animal experiments were approved by the Animal Care and Use Committee at the Research Institute for Microbial Diseases at Osaka University. [permit number: 2589, 3514].

### Data Availability
The sperm chain code data of morphology has been uploaded as a Supplemental File.

## Supplemental Information

Supplemental information for this article can be found online at http://dx.doi.org/10.7717/peerj.3913#supplemental-information.

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
