# Peer review of "Mouse spermatozoa with higher fertilization rates have thinner nuclei"

_PeerJ, doi:10.7717/peerj.3913_

## Round 0.1 · original submission · Minor Revisions

The topic of this manuscript is interesting and the results are potentially important for human reproduction. The manuscript however requires several minor changes proposed by reviewers 1 and 3.

Reviewer 1 ·

Basic reporting

The article is clear, comprehensible written. Despite the literature was well referenced and relevant, I consider that the author should refer to the 2010 version of The World Health Organization, and not to the one that was published in 1999.

Experimental design

no comment

Validity of the findings

no comment

Additional comments

The present study is an interesting work applying geometric morphometric to analyze sperm head contours within a population and between mouse strains and/or fertilization stages. The authors identified morphological indicators of highly fertile spermatozoa, and developed a pipeline that might also be applicable to human normal sperm. However, there are some aspects that the authors should take into consideration.

- The authors said that their method would greatly facilitate and enhance current reproductive technologies in an objective, high-throughput manner. It is not clear how this method can be used in the clinic laboratory to select a better sperm to perform ICSI. Authors should explain with more details this aspect.

- The percentage of sperm hyperactivation correlates with high fertilization rates and is different depending on the fertilization stage (cauda sperm vs PVS sperm). It is possible that the difference in fertilization rates between the 2 strains is due to differences in the percentage of hyperactivation. The author should also perform analyses between motility patterns and aspect ratio in BDF1 (in different fertilization stages) and B6N sperm populations.

- In Figure 4, there are only 3 animals analyzed, I recommend at least 4 to increase the number of cells per groups.

- It would be interesting if the authors perform what they proposed in the discussion: to test dithiothreitol on caput/cauda sperm and analyze head minor axis and aspect ratio of sperm. These results will explain the changes in these parameters during sperm maturation.

- Supplemental Figures: there is not any Figure legend and they are necessary to understand those figures.

- Supplemental Figure 2: It is not clear if they stimulated the acrosome reaction such as using progesterone or ZP or if the authors evaluated spontaneous acrosome reaction. It is possible that acrosome reaction by zona pellucida change the sperm aspect ratio.

- Line 97, authors wrote Fig.S4C, and there is not such panel in Fig.S4.

- Line 165, the authors should define also PC4, PC5.

Reviewer 2 ·

Basic reporting

This manuscript's premise is that the fertilizing potential of spermatozoa is correlated with their overall morphology. The authors focus in subtle differences between sperm, not in dramatic differences in morphology. The manuscript is well written and the assumptions made to compare fertilizing potential are sound. From the biology point of view, the results are interested and have the potential to be relevant in clinical practice. However, this reviewer's expertise in cell imaging is not sufficient to give an opinion on the imaging strategies as well as on the mathematics used to produce the data. I hope other reviewers of this manuscript are experts in cell imaging.

Experimental design

As said, assumptions made to choose sperm for analysis are sound. However, the conclusions heavily rely on the validity of the cell imaging methodologies used. Another reviewer should comment on this.

Validity of the findings

Nothing to add

Additional comments

Nothing to add

Reviewer 3 ·

Basic reporting

The authors presented a clear introduction with citations relevant to the topic and a clear research question.

1) In the line 83 of the introduction, the authors state that the sperm head contours of spermatozoa at different “fertilization stages” was analyzed. This is confusing to the reader, because the authors are using a broader definition of fertilization than the classical (the ability of a spermatozoa to penetrate and fuse with an egg (Yanagimachi, 1994)). In their definition, they are including sperm maturation in the epididymis and sperm capacitation in the female tract as part of fertilization (Okabe, 2013) instead of as required steps to acquire fertilizing-capacity. Therefore, it would be useful if the authors clarify this point by adding a sentence in which this broader definition of fertilization is stated.

Experimental design

The experimental design is not completely clear.

2) About the in vitro fertilization experiments:
a) Line 105: The authors state that the eggs were placed in TYH medium drops. Were the cumuls cells previously removed? If not, this line should state that “Cumulus-enclosed oocytes were placed in 100 µl drops of TYH medium”.

b) For this reviewer is not clear why the authors chose 8 hours of co-insemination between spermatozoa and oocytes before collecting sperm in the perivitelline space. It has been shown that after half an hour fertilization sperm has already fused with the oocyte (Tateno et al, 2013). In addition, Storey et al (1984) reported that mouse capacitated spermatozoa penetrated the zona of cumulus intact eggs in less than two hours after insemination. Therefore, after 8 hours of fertilization it seems difficult to find sperm in the perivitelline space unless fertilization and the cortical granule enzymatic reaction was inactivated to avoid polyspermy blockage and allow more spermatozoa to penetrate the zona. Is this the case? Please clarify this point.

As a minor comment, in the line 116: the world “eliminated” should be replaced by “removed”.

3) As this work is based in imaging analysis, it is important in the section of imaging and analysis to specify the camera and the settings that were used for taking the images. The specifications of the objective lens used are also important (for example, numerical aperture).

4) Minor comment. In the section of statistics, please state if when applying parametric tests, the assumptions of normality and homogeneity of variances were evaluated, and if any transformation of data was required.

Validity of the findings

The data is robust, statistically sound and controlled.

Some minor changes would improve the quality of the work.

5) Figure 4 a. The axis Fertilization stage is misleading because the graph is including different stages of maturation in the epididymis together with spermatozoa recovered from the female reproductive tract.

6) As a minor comment, in line 218 would be useful to disclose the fertilization rates of BDF1 and B6N cauda epididymis-isolated spermatozoa obtained in your laboratory.

Additional comments

The paper entitled “Mouse spermatozoa with higher fertilization rates have thinner nuclei” proposes a new way of morphometric and statistical analysis of mouse sperm heads as fertility indicator. The work presented combines imaging analysis with a multivariate statistics method that is intended to develop a new way of classifying sperm and determine fertility potential.
The data presented is good however the results section flow was not the best. The text guides the reader to start from supplementary figure 2 , then jumps to Figure 1 A, jumps again to supplementary figure 3, before going back to Figure1B and then continues with all the supplemental information before describing figure 2. I suggest to re-evaluate which information to include in the main figures and which one in the supplementary materials, in order to make the reading and understanding easier for the readers.
Finally, it is important to remark that the manuscript shows that the Principal Components Analysis was useful to identify the main variables that represent the morphological variability of the sperm head nucleus: width, and the hook shape of the tip.

---

## Round 0.2 · accepted · Accept

I believe that the results presented in this manuscript are important not only for reproduction field, but also for general scientific community with potential application outcomes.

Reviewer 1 ·

Basic reporting

no comment

Experimental design

no comment

Validity of the findings

no comment

Additional comments

The authors took care to answer to my questions and make this revised version of the manuscript easier to follow. I apologize for not finding the supplementary figure legends in the first round of revision.

Reviewer 2 ·

Basic reporting

The authors have addressed all comments from reviewers.

Experimental design

No Comment.

Validity of the findings

No comment.

Additional comments

The authors have addressed all comments.

Reviewer 3 ·

Basic reporting

The authors have carefully addressed the issues that were pointed out by this reviewer. The revised version of the manuscript is improved and no further corrections are needed.

Experimental design

No further comments.

Validity of the findings

No further comments

Additional comments

No further comments